# Interaction of Very Large Scale Motion of Coherent Structures with Sediment Particle Exposure

**Sencer Yücesan** [1,2,*], **Daniel Wildt** [1,2], **Philipp Gmeiner** [1,2], **Johannes Schobesberger** [1,2], **Christoph Hauer** [1,2], **Christine Sindelar** [2], **Helmut Habersack** [2] and **Michael Tritthart** [2]

1   Christian Doppler Laboratory for Sediment Research and Management, Muthgasse 107, 1180 Vienna, Austria; daniel.wildt@boku.ac.at (D.W.); philipp.gmeiner@boku.ac.at (P.G.); johannes.schobesberger@boku.ac.at (J.S.); christoph.hauer@boku.ac.at (C.H.)

2   Institute of Hydraulic Engineering and River Research (IWA), University of Natural Resources and Life Sciences, Muthgasse 107, 1180 Vienna, Austria; christine.sindelar@boku.ac.at (C.S.); helmut.habersack@boku.ac.at (H.H.); michael.tritthart@boku.ac.at (M.T.)

*   Correspondence: s.yuecesan@boku.ac.at

**Abstract:** A systematic variation of the exposure level of a spherical particle in an array of multiple spheres in a high Reynolds number turbulent open-channel flow regime was investigated while using the Large Eddy Simulation method. Our numerical study analysed hydrodynamic conditions of a sediment particle based on three different channel configurations, from full exposure to zero exposure level. Premultiplied spectrum analysis revealed that the effect of very-large-scale motion of coherent structures on the lift force on a fully exposed particle resulted in a bi-modal distribution with a weak low wave number and a local maximum of a high wave number. Lower exposure levels were found to exhibit a uni-modal distribution.

**Keywords:** coherent structures; hairpin-vortex packets; hydrodynamic forces; particle entrainment; very-large-scale motions; turbulent channel flow





## 1. Introduction

Coherent structures in the vicinity of the wall region in turbulent flow have received significant attention after the pioneering findings of hairpin-like vortices [1] that possess spatial coherence. Boundary layer studies found that the streamwise elongated coherent structures carry intense turbulent kinetic energy (TKE) in the near wall region [2]. Velocity streaks are the primary examples of this. Kline et al. [3] described the motion of low-speed fluid flow from the viscous sublayer to the outer portions of the inner region as ejections, which are responsible for the production of TKE, whereas sweeps are responsible for the movement of high speed fluid flow towards the viscous sublayer. This quasi-cyclic organized motion of fluid flow is referred to as bursting [4]. Furthermore, it was indicated that the large-scale motion (LSM) of three-dimensional coherent structures resides within the buffer and the logarithmic layer [4]. From this resulted an early classification of coherent structures, which were limited to wall-bounded low Reynolds number flows, i.e., a scale of size $\mathcal{O}(\delta)$, where $\delta$ is the boundary layer thickness [4].

Later advances revealed the existence of two different scales in turbulent flows. These are LSM and very-large-scale motion (VLSM) of coherent structures [5]. Kim et al. described that VLSM are gathered from small hairpin packets to form long packets that are spatially much longer than the LSM. They found that a bi-modal distribution in the spectrum analysis of velocity fluctuations results in two separate wavelengths that correspond to VLSM and LSM. Since then, VLSM has been investigated in pipe flows [5,6], turbulent flows [7–11], and open channel flows [12–17].

Hydrodynamic forces, in particular drag and lift, acting on a sediment particle in turbulent flows have been extensively studied through experiments [18–27] and numerical

simulations [28–34]. Recent investigations applied vortex-detection methods in order to identify the coherent structures that are responsible for the generation of hydrodynamic forces. Chan-Braun et al. [30] reported that streamwise elongated velocity streaks in the buffer-layer are the responsible mechanisms for the correlation between the streamwise velocity fluctuations and drag force. Vowinckel et al. [31] studied the entrainment of a sediment particle over a fixed bed of a square arragement of spheres. They reported the existence of streamwise aligned, counter-rotating structures at the onset of particle entrainment. While these two studies adressed the influence of coherent structures on the hydrodynamic forces of particles, their study was limited to velocity streaks within the buffer layer. Schobesberger et al. [35] performed an experimental investigation of a sediment particle resting on a smooth-wall and identified the passing LSM of coherent structures at the onset of particle entrainment. A complementary numerical study of this experiment, investigating a single sediment particle fully exposed to turbulent open channel flow on a smooth-wall, was performed by Yücesan et al. [34], in the following referred to as the $M2$ case. They found out that vortices that were characterised by their spatial scale (in particular, in the wall-normal direction) to be similar or larger than that of the sediment particle produced a significant simultaneous increase of lift and drag forces, while interactions with small scale coherent structures resulted in negligible changes in hydrodynamic forces.

Early investigations of the drag force while using spectrum analysis reported that low frequency streamwise velocity fluctuations produce a quasi-steady drag force [36]. In addition to that, high frequency fluctuations were found to amplify the nonlinearity due to high order velocity fluctuations in the spectrum curve, which is highly dependant on the exposure level of the particle. Experimental studies of Cameron et al. [15,17] were the first to analyse the effect of VLSM on the drag force of a spherical roughness element. Their premultiplied spectrum analysis of the drag force was characterised with two modes of scales which were low-frequency and high-frequency peaks. Low frequency peaks were characterised by the influence of VLSM, whereas high frequency peaks were addressed to the influence of the pressure field in the vicinity of the particle. Their study also reported that the magnitude of the spectral peaks increases with increasing particle exposure and channel depth. The premultiplied spectrum of the drag force of a fully hidden particle was not affected by the VLSM due to a shielding effect which was also identified by Dwivedi et al. [36]. While Chan-Braun et al. [30] were the first to identify a bi-modal distribution of scales in the spectrum analysis of the drag force, their study evaluated neither VLSM nor the effect of particle exposure.

Lift force is the less understood component of the hydrodynamic forces on a sediment particle compared to the drag force. Recent investigations reported that lift force is poorly correlated with the streamwise and vertical velocities and vertical momentum flux based on the measurements at the upstream side as well as at the top side of the particle [22]. Smart & Habersack [21] reported that pressure difference above and beneath the particle is large enough to entrain the particle. Dwivedi et al. [25] noted that increasing particle exposure resulted in an increase of the skewness of the lift force. Their study also reported co-spectra of the lift force and the flow field. Furthermore, a spectrum analysis of the lift force was reported to exhibit two scaling ranges [37].

Today, our understanding of the interrelation between coherent structures and the drag force has been established through varying particle exposure, auto- and cross-correlation of the flow field, high order statistics and spectrum analysis. However, observations on the lift force acting on the sediment particles were only limited to time series of pressure measurements. Therefore, the effect of LSM and VLSM on the lift force still remains widely unknown. The aim of the present study is to perform numerical simulations in order to analyse the effect of LSM and VLSM on the lift force. The numerical simulation employs two different configurations of a rough boundary, which are denoted as $SP1$ and $SP2$ cases, in order to study the effect of roughness elements on the hydrodynamic forces. A comparison is performed with the aforementioned $M2$ case, in which a particle on a smooth-wall

is fully exposed to the flow [34]. Auto and Cross-correlation of the hydrodynamic forces, in particular the lift force, with the flow field has been evaluated. The spectra and the premultiplied spectra of the velocity fluctuations as well as the hydrodynamic forces acting on the particle are presented and discussed.

## 2. Numerical Methods

### 2.1. Domain and Boundary Conditions

A systematic variation of the exposure level of a spherical sediment particle of fixed size $d = 0.026$ m was performed, placed in an array of spherical roughness elements and exposed to turbulent flow with a high Reynolds number. The numerical simulation consists of two configurations of the bottom wall with different sizes of the roughness elements in a square arrangement. The diameter of the roughness elements in the $SP1$ and $SP2$ cases is characterised as $d/2$ and $d$, respectively. The particles were separated from each other by a distance of $\Delta p = 0.154d$, where $\Delta p$ is the shortest distance between the particles. The ratio of the roughness elements to the channel height is characterised by $d/h = 0.076$ in the $SP1$ case and $d/h = 0.152$ in the $SP2$ case, where $h$ is the channel height. The selection of the particle diameter was based on the entrainment conditions of a single sediment particle exposed to fully developed turbulent open channel flow as described in the experimental study by Schobesberger et al. [35]. The no-slip/no-penetration $u_i = 0$ boundary condition was applied on the sediment particle surface. A schematic is presented in Figure 1 to illustrate the setup of the simulation in the $SP1$ and $SP2$ cases. An open channel flow (OCF) was considered with the dimensions of $0.9 \times 0.171 \times 0.3$ m. The streamwise, wall-normal and spanwise directions in a Cartesian coordinate system were denoted by $x$, $y$ and $z$, respectively. Throughout the manuscript $\langle . \rangle$ indicates time averaging.

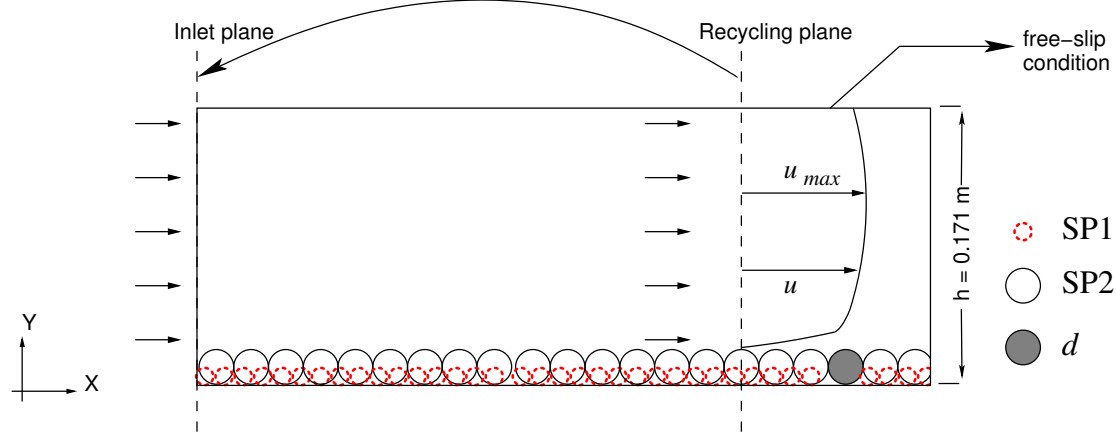

**Figure 1.** Principle sketch of the flow field in the concurrent simulation for the $SP1$ and $SP2$ cases.

A concurrent simulation method was employed to compute a fully developed turbulent open channel velocity profile in the streamwise direction [38,39]. The method uses a plane orthogonal to the flow direction located in the downstream region of the channel to sample the instantaneous velocity and pressure field which is then used as an inlet condition. Therefore, the approach flow in the simulation is characterised as a fully developed rough-wall turbulent open channel velocity profile.

The simulation employed a no-slip/no-penetration $u_i = 0$ boundary condition at the bottom wall, the lateral walls as well as for the roughness elements, $\partial u/\partial y = \partial w/\partial y = 0$, $v = 0$ at the free surface and $\partial u_i/\partial x = 0$ at the outlet, where $u, v$ and $w$ are the streamwise, wall-normal and spanwise velocities, respectively. The outlet is located $\approx 5.8d$ (or $\approx 7308$ $(\nu/u_\tau)$ in viscous units) downstream of the center of volume of the particle. While this is expected to be large enough to avoid any influence of the outlet boundary condition on the hydrodynamic forces acting on the particle, a further downstream extension of the domain would have been desirable, but proved computationally unfeasible. The do-

main of the simulation was discretized with a fully structured mesh by $24.24 \times 10^6$ and $19.17 \times 10^6$ cells in the $SP1$ and $SP2$ cases, respectively. The boundary layer thickness $\delta$ was calculated based on the maximum mean streamwise velocity $u(y)_{max}$ in the wall-normal direction at the top of the spherical sediment particle. Similarly, the friction velocity $u_\tau$ has been estimated based on the extrapolation of the linear segment of the total stress curve. The streamwise length of the recycling plane was chosen according to the analysis of the two-point correlation (TPC) of velocity fluctuations along the streamwise direction depicted in Figure 2. The correlations were observed to drop to zero $\approx 5d$ away from the inlet for the $SP1$ and $SP2$ cases.

The grid densities near the sediment particle in the present simulations were calculated for the $SP1$ ($s^+ \approx 1$) and $SP2$ ($s^+ \approx 1.2$) cases. Additionally, the position of the first grid node from the wall was set to $y^+ \approx 1.5$ and $y^+ \approx 1.7$ for the $SP1$ and $SP2$ cases, respectively, which is well below ten wall units. Positions of the grid nodes within the domain in all other directions are within 25 wall units. Therefore, the grid resolution is expected to be fine enough to resolve most of the energy within the channel. Details of the simulation parameters, including the $M2$ case [34], are summarized in Table 1.

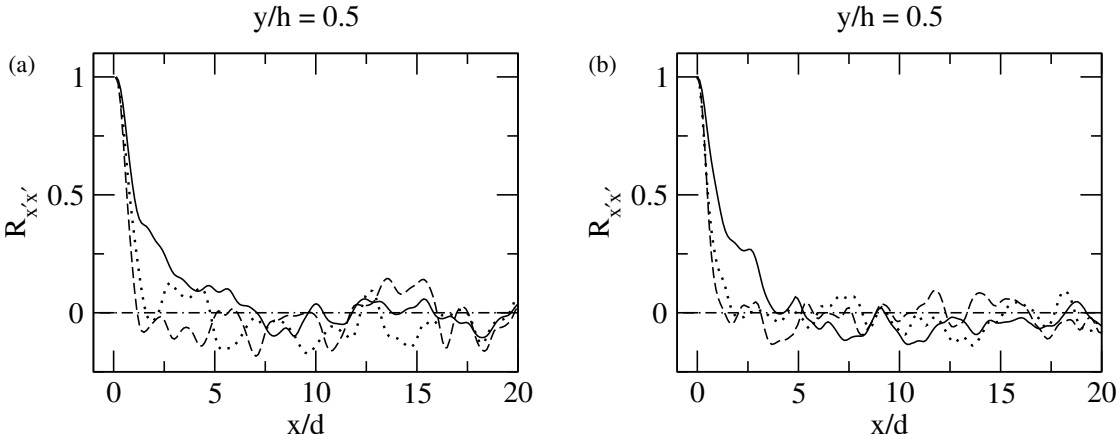

**Figure 2.** TPC of velocity fluctuations ($u'$, $v'$, $w'$ presented with $(-)$, $(--)$, $(\bullet)$, respectively) at the center of the bulk along streamwise ($R_{x'x'}$) direction: (**a**) $SP1$; (**b**) $SP2$ case.

**Table 1.** Simulation parameters: $U_b = \frac{1}{h}\int_0^h \langle u \rangle dy$ is the spatially and temporally averaged bulk velocity, $u_\tau = \sqrt{\langle \tau_w \rangle / \rho}$ the spatially averaged friction velocity at the bottom wall, $Re_b = hU_b/\nu$ is the bulk Reynolds number, $Re_\tau = hu_\tau/\nu$ is the friction Reynolds number, $\delta$ is the boundary layer thickness, $\Delta x^+, \Delta y^+, \Delta z^+$ and $\Delta s^+$ are the grid spacing along streamwise, wall-normal, spanwise and sediment particle in viscous units, respectively, $T^+ = TU_b/h$ is total simulation time in which the statistical information was collected without taking into account turbulent transition.

| Case | $U_b/u_\tau$ | $Re_b$ | $Re_\tau$ | $\delta$ | $\Delta x^+$ | $\Delta y^+$ | $\Delta z^+$ | $\Delta s^+$ | $T^+$ |
|------|-------------|--------|-----------|----------|--------------|--------------|--------------|--------------|-------|
| $M2$ | 23.25 | 71,755 | 3443 | $0.585h$ | 15.5 | 7.7 | 15.5 | 1.9 | 125 |
| $SP1$ | 9.10 | 75,327 | 8284 | $0.666h$ | 16.5 | 1.5 | 16.5 | 1 | 79.3 |
| $SP2$ | 7.42 | 77,096 | 10,373 | $0.737h$ | 24.5 | 1.7 | 24.5 | 1.2 | 94 |

### 2.2. Methodology and Turbulence Statistics

Large Eddy Simulation (LES) of the unsteady Navier-Stokes and continuity equations was performed to compute the incompressible, Eulerian flow field. The numerical setup of the simulation is identical to Yücesan et al. [34]. The OpenFOAM [40] open-source software package has been used for the numerical simulations. Convective terms were discretized using an upwind-biased method, gradient terms by a central differencing scheme. The time derivative was discretized by an implicit backward differencing method. In order to preserve temporal accuracy, the Courant number was kept $\leq 0.5$. The accuracy of the numerical schemes in time and space is of second order.

The time-averaged velocity field was gathered based on plane-averaged velocity data sampled at $x/d \approx 28.27$. The effective flow height was calculated as $h_{eff} = h - d_{eff}$, where $d_{eff}$ is the artificial position of the wall defined as $d_{eff} = 0.8d/2$ and $d_{eff} = 0.8d$ for the $SP1$ and $SP2$ cases, respectively. Figure 3a depicts the mean streamwise velocity normalized by the bulk velocity. The resulting velocity profiles show that the maximum velocities occurred below the water surface, indicating the presence of secondary currents (SC) in the channel [41]. Secondary currents can be a significant mechanism of delivery of momentum from and towards the channel boundaries if the aspect ratio (width to depth) is lower than a certain value. Despite the influence of the SC in our numerical study, we have omitted their effect as the scope of the present manuscript is not the interrelation between VLSM and SC. The root mean square of velocity fluctuations for the $SP1$ and $SP2$ cases are presented in Figure 3b and compared to the smooth-wall case $M2$. A visualization of the mean velocity magnitude of the flow field is depicted in Figure 4.

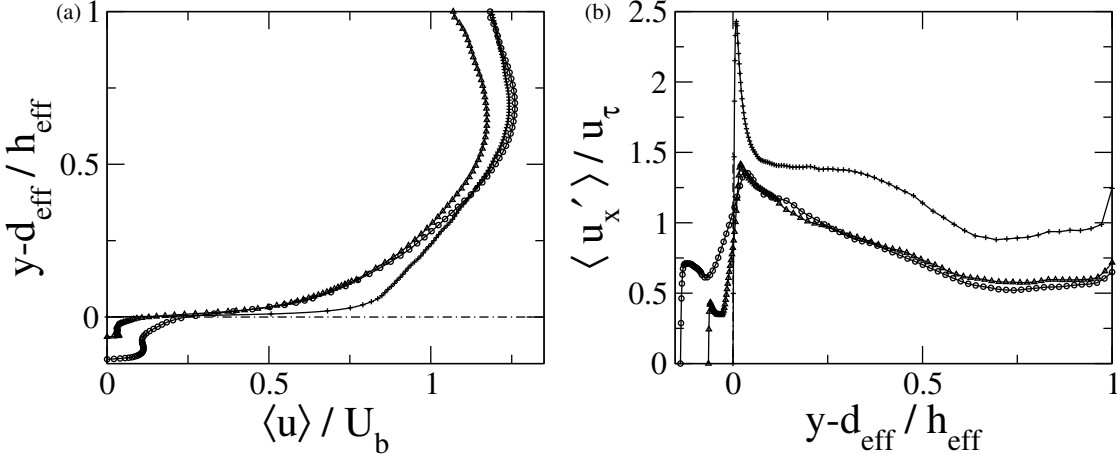

**Figure 3.** Normalized mean streamwise velocity for the $SP1$ ($\triangle$) and $SP2$ ($\circ$) cases, compared to the mean centerline velocity profile for $M2$ ($+$) (**a**); root mean square of velocity fluctuations for the $SP1$ ($\triangle$) and $SP2$ ($\circ$) cases, compared with the smooth-wall case $M2$ ($+$) (**b**).

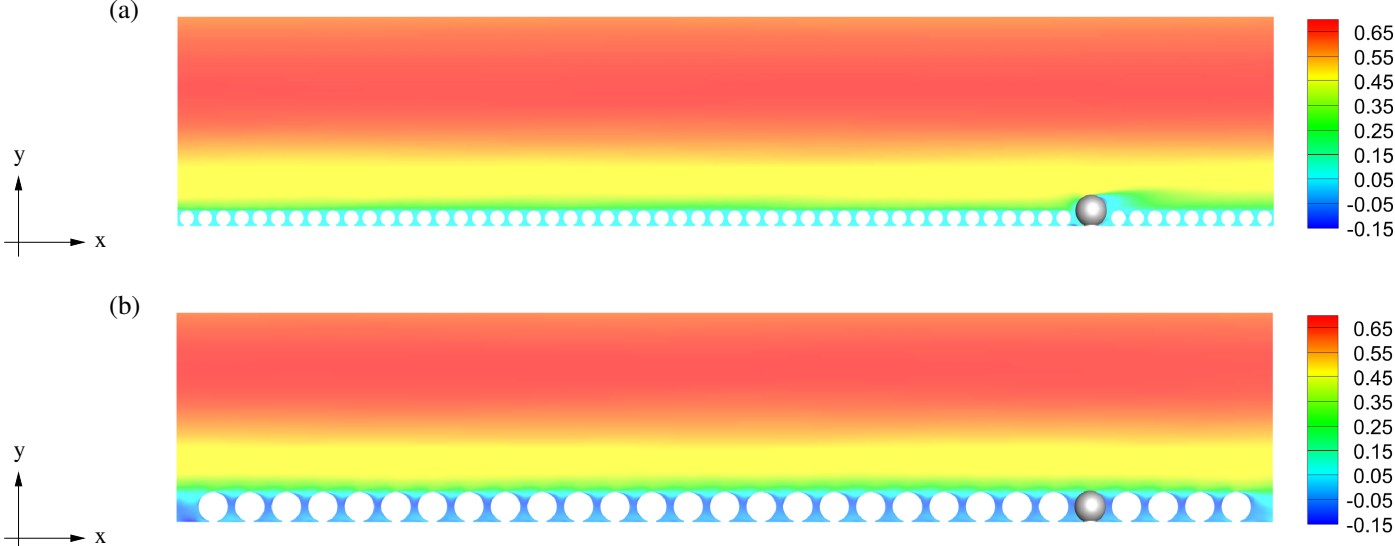

**Figure 4.** Streamwise aligned cross-sectional plane for the $SP1$ (**a**) and $SP2$ (**b**) cases. The mean streamwise velocity of the flow field presented with colour contours.

## 3. Results and Discussion

### 3.1. Hydrodynamic Forces Acting on the Spherical Particle

Hydrodynamic forces acting on the spherical sediment particle are defined, as follows,

$$\boldsymbol{F} = -\int_s p\boldsymbol{n}\,ds + \int_s \boldsymbol{\tau} \cdot \boldsymbol{n}\,ds \tag{1}$$

where $\boldsymbol{F}$ is the total surface force that is acting on the target spherical particle, $\boldsymbol{n}$ is the surface normal vector, $s$ is the surface of the particle, $p$ is the kinematic pressure, and $\boldsymbol{\tau}$ is the stress tensor.

The computation of drag and lift coefficients is based on the hydrodynamic force formulation,

$$F_{\{D,L,Z\}} = \frac{1}{2} U_b^2\,A\,C_{\{D,L,Z\}} \tag{2}$$

where $U_b$ is the bulk velocity, $A = \pi D^2/4$ is the planform reference area of the spherical sediment particle, $F_D$ is the drag force, $F_L$ is the lift force, $F_Z$ is the lateral force, $C_D$ is the drag coefficient, $C_L$ is the lift coefficient, and $C_Z$ is the lateral force coefficient. Table 2 summarizes the statistics of the hydrodynamic forces, including the $M2$ case [34] in order to present the variation of the forces.

**Table 2.** Statistics of forces acting on the stationary sediment particle: $\sigma_D/\langle F_D\rangle$ is the standard deviation of the instantaneous drag force normalized by the mean drag, $\sigma_L/\langle F_L\rangle$ is the standard deviation of the instantaneous lift force normalized by the mean lift, $F_D^+ = \langle F_D\rangle/\rho v^2$ is the mean drag force presented in dimensionless form, where $\rho$ is the density of the fluid, $F_L^+ = \langle F_L\rangle/\rho v^2$ is the mean lift force presented in dimensionless form, $S_{D,L,Z}$ and $F_{L_{D,L,Z}}$ are the skewness and flatness of the regarding instantaneous forces, respectively.

| Case | $\sigma_D/\langle F_D\rangle$ | $\sigma_L/\langle F_L\rangle$ | $F_D^+$ ($\times 10^{-6}$) | $F_L^+$ ($\times 10^{-6}$) | $S_D$ | $S_L$ | $S_Z$ | $F_{L_D}$ | $F_{L_L}$ | $F_{L_Z}$ |
|------|------|------|------|------|------|------|------|------|------|------|
| $M2$ | 0.094 | 0.284 | 19.5 | 4.98 | 0.138 | 0.119 | 0.037 | 2.624 | 2.91 | 2.899 |
| $SP1$ | 0.347 | 0.47 | 10.3 | 4.1 | 0.441 | 0.262 | $-0.016$ | 3.315 | 3.361 | 3.321 |
| $SP2$ | 0.912 | 1.15 | 4.22 | 2.49 | 0.002 | 0.462 | $-0.249$ | 3.399 | 4.134 | 3.723 |

The mean drag coefficients $\langle C_D\rangle$, as shown in Figure 5, corresponding to the $SP1$ and $SP2$ cases, were identified as 0.211 and 0.082, respectively. The increase of the drag coefficient is associated with the exposed area of the particle to turbulent open channel flow. The coefficient of variation of the drag force $\sigma_D/\langle F_D\rangle$ increases with decreasing particle exposure. The $SP2$ case yielded a ratio of 0.912, whereas the $SP1$ case resulted in a significantly lower ratio of 0.347. The standard deviation of the drag force was observed to decrease with increasing particle exposure. However, the rate of change in the drag force is more pronounced than that of the variation of the standard deviation; therefore, the coefficient of variation approached $\approx 1$ for a fully hidden particle in $SP2$. The investigations of Schmeeckle et al. [22] show that the coefficient of variation of the drag force for a fully hidden particle becomes unity, whereas increasing exposure yielded a lower ratio, which is in line with our findings. High order statistics showed that the skewness of the drag is small for the $SP1$ and $SP2$ cases. The flatness of the drag forces exhibits a decrease with increasing exposure level. The half-exposed and fully exposed particles yielded similar flatness coefficients $\approx 0.33$ as compared to the $M2$ case.

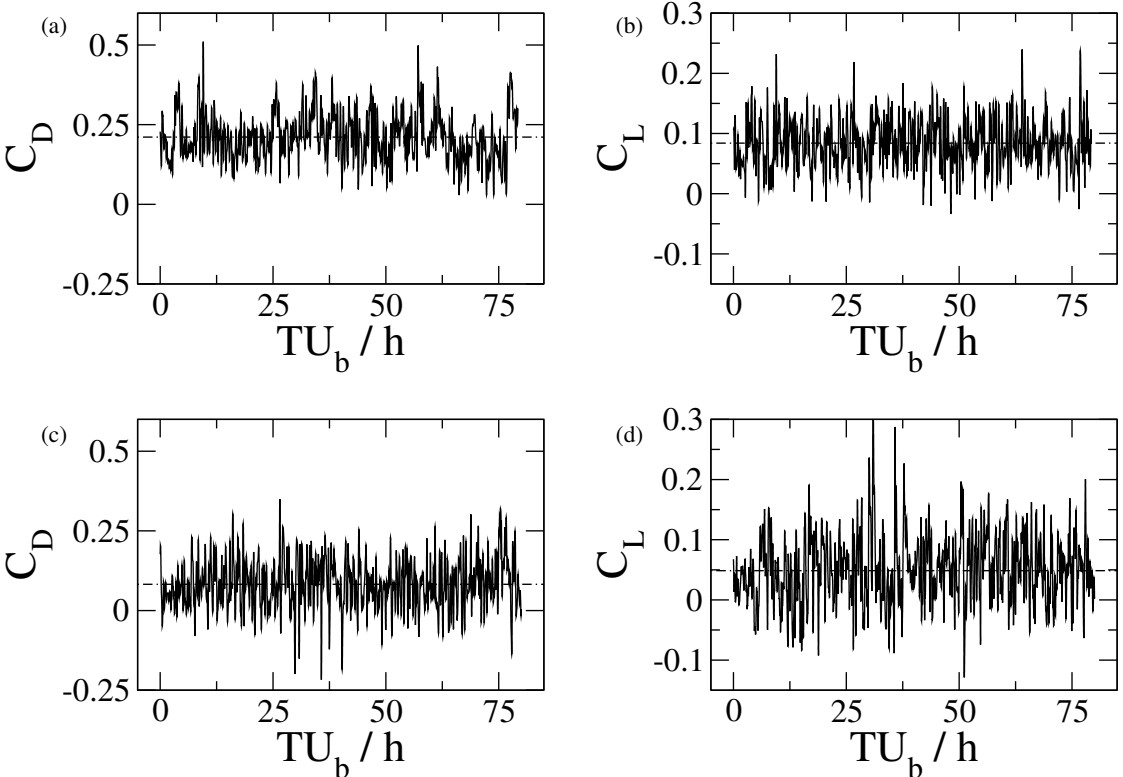

**Figure 5.** Time series of the drag and lift coefficients for the *SP*1 (**a**,**b**) and *SP*2 (**c**,**d**) cases.

Table 2 presents the variation of the mean lift force with respect to the particle exposure. Our results show that the mean lift force increases with increasing particle exposure. However, the standard deviation of the fluctuations of the lift force exhibited a decreasing trend with higher particle exposure. Earlier investigations conducted by Zeng et al. [42] in a low Reynolds number channel flow reported an increase of the lift coefficient with decreasing distance from the smooth-wall. The lift force on a rough boundary with varying elevation level was investigated by Schmeeckle et al. [22], who reported a decrease of the mean lift force with increasing distance of the particle from the bottom wall, simultaneously increasing exposure. Thus, it was observed that a particle residing on the bottom wall without an elevation increase results in an increase of the lift force with rising exposure. On the other hand, only increasing the elevation level with or without roughness elements yields a decrease of the positive pressure force generated at the bottom part of the particle. A possible explanation of this behaviour is that increasing the particle exposure results in an increase of the stagnation pressure at the leading edge, which, in turn, increases the lift force with a higher exposure level. High order statistics (i.e., skewness and flatness) of the lift force exhibit an increasing trend with decreasing exposure. The skewness of the lift force in the *SP*2 case was observed as $F_{L_L} = 0.462$, while increased exposure (i.e., *SP*1) yielded $F_{L_L} = 0.262$. The flatness of the *M*2 case exhibits a nearly Gaussian distribution, whereas the *SP*1 and *SP*2 cases resulted in a higher flatness coefficient.

### 3.2. Auto-Correlation Function of Lift and Drag Forces

Figure 6 shows the Auto-Correlation Functions (ACF) of the hydrodynamic forces on the spherical sediment particle for the *SP*1 and *SP*2 cases. The time lag was normalized with outer scaling.

Auto-correlation of the fluctuations of the drag force was observed to decay faster at a smaller lag with decreasing exposure of the particle. Auto-correlation of the drag in the *SP*2 case drops to the zero axis more rapidly when compared to the half-exposed particle in the *SP*1 case, as visible from Figure 6a. The findings of Yücesan et al. [34] also confirm the decaying trend, as a fully exposed particle exhibits even a higher correlation

over a larger lag. A possible explanation of this systematic decaying trend is associated with the streamwise elongated coherent structures, which results in a high auto-correlation over a large lag. A similar finding by Amir et al. [37] concluded that the cross-correlation of two particles positioned along the flow direction with a distance apart is higher with respect to the particles that are positioned spanwise, which is in line with our findings. ACF of the drag force in the $SP2$ case exhibits a local minimum of $-0.083$ at $TU_b/h = 0.3$, whereas the $SP1$ case does not exhibit any local minimum. A possible reason of the local minimum, observed only in the $SP2$ case, was assumed to be the effect of pressure forces, in particular fluctuations of the pressure field near the bottom wall, according to previous studies [37,43].

ACF of the fluctuations of the lift force for the $SP1$ and $SP2$ cases in Figure 6b showed a close promixity of decay over lag when compared to the auto-correlation of drag forces. Therefore, the auto-correlation of the lift forces is less likely to be influenced by the varying particle exposure in the investigated setup. This explains that streamwise elongated coherent structures do not affect the lift force over a larger lag. In Figure 6b, it can be seen that the $SP1$ case does not result in a local minimum, instead drops to the zero axis at $TU_b/h \approx 0.25$ and fluctuates. A comparison of the auto-correlation between fluctuations of the drag and lift forces in the $SP1$ case indicates that the lift force more rapidly drops to the zero axis, whereas the ACF of the lift force in the $SP2$ case exhibits a drop over a slightly longer time.

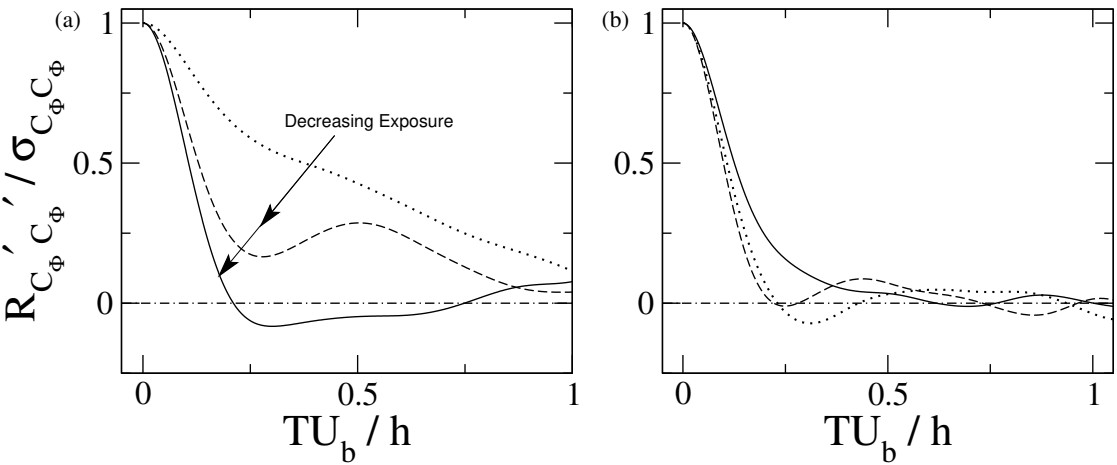

**Figure 6.** Auto-correlation functions of drag (**a**) and lift (**b**) coefficients for the $SP2$ (−) and $SP1$ (−−) cases, when compared with the literature results of the $M2$ case in Yücesan et al. [34] ($\cdots$).

### 3.3. Cross-Correlation Function of Lift and Drag Forces

Cross-correlation functions (CCF) of the drag and lift forces, as well as their correlation with respect to the flow properties, in particular streamwise and wall-normal velocity fluctuations, were analysed in order to understand how flow structures relate to the hydrodynamic forces in the vicinity of the particle. The flow parameters were sampled at two specific locations to assess the influence of the streamwise position on the correlation coefficients. The wall-normal position was kept constant at $y = 1.15d$, whereas two different streamwise positions were selected: at one spherical diameter $(d)$ upstream of the particle $(x/d = -1)$ and at the top of the particle at $(x/d = 0)$.

Figure 7 presents the CCF between drag and lift coefficients of the particle in the $SP1$ and $SP2$ cases and it provides a comparison with respect to the $M2$ case of Yücesan et al. [34]. In general, positive correlation indicates that fluctuations of the drag and lift forces have the same positive sign, whereas the minimum peaks indicate the opposite signs of fluctuating values [32]. Fluctuations of the drag and lift forces for the $SP1$ and $SP2$ cases at zero lag $TU_b/h = 0$ are weakly correlated: $R_{C_D'C_L'}/\sigma_{C_D}\sigma_{C_L} = (-0.055, -0.042)$. The $SP2$ case exhibits local minimum and maximum correlation coefficients of $(-0.363, 0.249)$ at

$TU_b/h = (-0.114, 0.11)$ with a separation time between the local maxima and minima $\Delta TU_b/h = 0.224$, whereas the $SP1$ case results in local maximum and minimum coefficients of $(-0.265, 0.216)$, with a separation time of $\Delta TU_b/h = 0.21$. Therefore, the time lags for which the maximum and minimum peaks are observed, are almost identical in both cases. The $SP2$ case exhibits a slightly higher lag of $\approx +0.014$ between the peak points of its correlation when compared to the $SP1$ case. A possible explanation of this behaviour is that bulk velocity in $SP2$ is higher as compared to the $SP1$ case, thus the wavelength of the correlation in the $SP2$ case is larger. The correlation coefficient of both cases indicates that the $SP2$ case is uncorrelated over a larger lag. However, the half-exposed particle in case $SP1$ results in weak fluctuations of correlation. The amplitude of the negative correlation coefficient is significantly reduced in the $SP1$ case, when compared to $SP2$, as also visible from Figure 7. However, the gap between the maximum positive peaks of the correlation coefficient of these two cases remains less affected with a slightly higher positive correlation of the $SP2$ case. Thus, the fully hidden particle in case $SP2$ results in a higher amplitude of the correlation when compared to the half-exposed particle in $SP1$.

The correlation coefficient between the drag and lift forces in previous studies on a rough-wall [30,32,44,45] was reported to have a local maximum at $\approx (0.23 - 0.55)$, and a local minimum at $\approx -0.5$ for a single particle, which is in line with our findings. A similar investigation [37] studied the correlation of two adjacent particles and a maximum correlation coefficient of 0.115 was reported. The most surprising result reported by [34] is the CCF between fluctuations of the drag and lift forces of a single sediment particle resting on a smooth-wall, which does not exhibit a local minimum, but two maximum peaks, which indicates that drag and lift forces result in a positively fluctuating correlation.

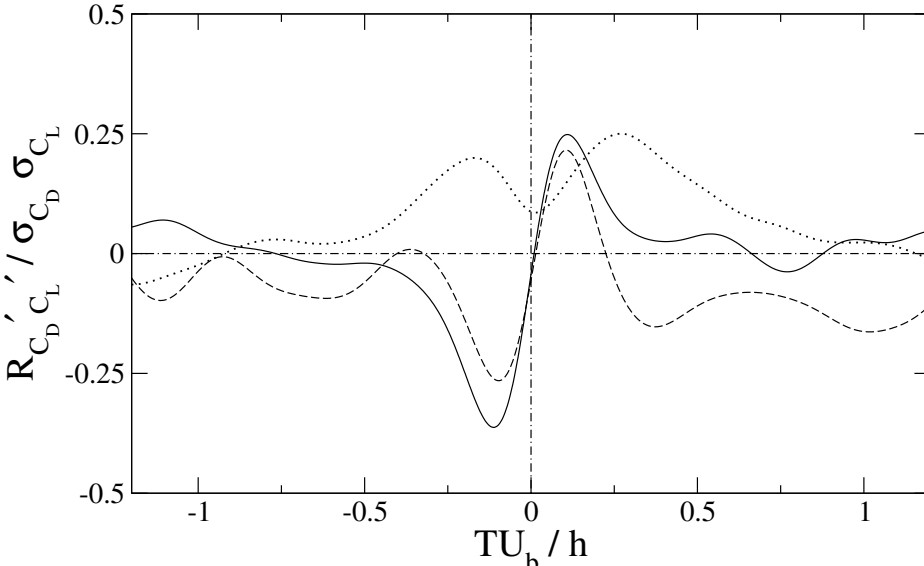

**Figure 7.** Cross-correlation function of the fluctuations of the drag and lift forces for the $SP2$ $(-)$ and $SP1$ $(--)$ cases, compared with the literature results of the $M2$ case in Yücesan et al. [34] $(\cdots)$.

Table 3 provides the minimum and maximum values of the correlation coefficients of the hydrodynamic forces $(C_D, C_L)$ with respect to the flow properties $(u', v')$. The samples of the velocity fluctuations were taken at two separate streamwise positions, in particular one particle diameter upstream $(x/d = -1)$ and at the top of the particle $(x/d = 0)$. Our results show that all of the presented cases exhibit a weak correlation, although the correlation coefficients in the $SP1$ case exhibit slightly more pronounced values when compared to the $SP2$ case. Thus, a correlation of the flow variables and hydrodynamic forces was considered to be neglible, which corresponds to the findings of [22].

**Table 3.** Cross-correlation of the streamwise and wall-normal velocity fluctuations at two separate locations with respect to the hydrodynamic forces.

| | $x/d = -1$ | | | $x/d = 0$ | | |
|---|---|---|---|---|---|---|
| **Case** | $R_{u'C_D'}/\sigma_{u'C_D'}$ | $R_{u'C_L'}/\sigma_{u'C_L'}$ | $R_{v'C_L'}/\sigma_{v'C_L'}$ | $R_{u'C_D'}/\sigma_{u'C_D'}$ | $R_{u'C_L'}/\sigma_{u'C_L'}$ | $R_{v'C_L'}/\sigma_{v'C_L'}$ |
| *SP*1 | $(-0.174, 0.234)$ | $(-0.100, 0.117)$ | $(-0.142, 0.090)$ | $(-0.182, 0.241)$ | $(-0.097, 0.104)$ | $(-0.106, 0.097)$ |
| *SP*2 | $(-0.099, 0.110)$ | $(-0.101, 0.078)$ | $(-0.095, 0.095)$ | $(-0.108, 0.106)$ | $(-0.101, 0.068)$ | $(-0.107, 0.086)$ |

### 3.4. Spectrum Analysis

The power spectrum and premultiplied power spectrum of the streamwise velocity fluctuations as well as fluctuations of the drag and lift forces acting on the sediment particle were computed. The spectrum analysis of the velocity fluctuations was performed based on samples that were taken at the top of the spherical particle ($x/d = 0$) at a distance $y = 1.15d$ away from the bottom wall. Premultiplied spectra of the streamwise velocity fluctuations are used in order to identify the presence of the very-large-scale motion of the coherent structures and their respective wavelengths. The very-large-scale and large-scale coherent structures are corresponding to low frequency and high frequency fluctuations, respectively. They were identified in many studies in turbulent channel flows [8,9], turbulent pipe flows [5,6], and rough-bed channel flows [14,15,17]. Figure 8 provides a visual impression of the passing vortices in terms of their size and spatial development. Clockwise and counter-clockwise rotating vortices are both extending up to the channel height ($y = h$).

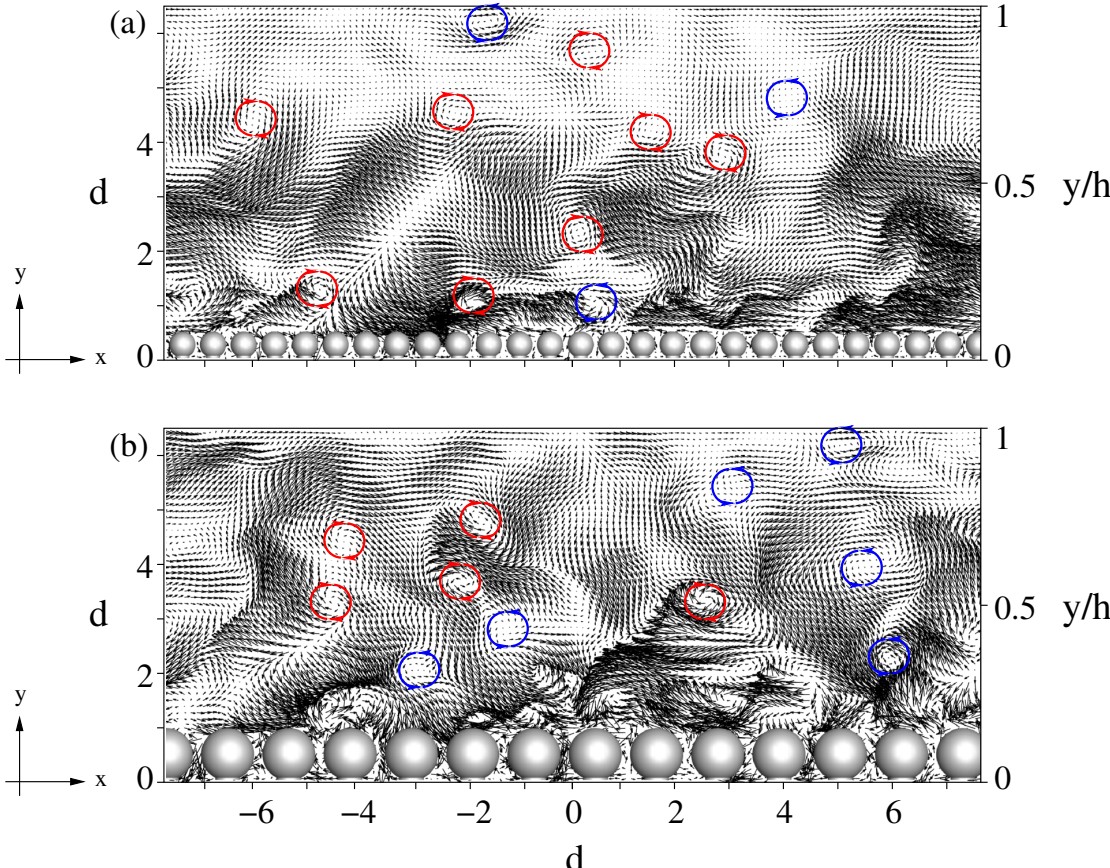

**Figure 8.** Streamwise aligned cross-sectional plane for the *SP*1 (**a**) and *SP*2 (**b**) cases, showing the instantaneous velocity fluctuations. Vortices rotating clockwise (red) and counter-clockwise (blue) are indicated, extending up to the channel height ($y = h$).

The visualization of the energy spectrum corresponding to the velocity field was performed by applying the discrete Fourier transformation of the whole time-series of the streamwise velocity fluctuations. A windowing operation of the signal was not applied, due to very large size of the time-series, which is constituted of $N = 44{,}721$ and $N = 61{,}894$ samples for the $SP1$ and $SP2$ cases, respectively. The wave number of the spectra was calculated based on Taylor's frozen turbulence hypothesis of $k_x = 2\pi f / u_{b_{(M2,SP1,SP2)}}$ with the assumption that convection velocity is equal to the mean bulk velocity. Although some studies [5,14] have reported the deficiency of Taylor's hypothesis, the convection velocities that were reported in the literature based on a rough-wall [32,37] were found to be within the range of $(0.66\text{–}0.72)U_b$. Therefore, a higher wave number would be expected in the determination of the true wave number than in our assumption. Thus, we consider an underestimated wave number to actually strengthen our results.

Figure 9a–f show the results of the energy spectrum analysis, corresponding to the $M2$, $SP1$, and $SP2$ cases, respectively. Figure 9a,c,e depict the energy spectra of the velocity fluctuations. The trend of the energy spectrum in all cases exhibits a slight increase within the low wave number region until a local maximum was obtained. Despite the high amplitude fluctuations present in the spectra of the signal, a clear trend of the power law $(k^{-1})$ is visible, which separates the low wave number and high wave number regions. Kim et al. [5] interpreted the beginning of the power law $k^{-1}$ region as an indication of the low wave number mode. Based on the beginning of the power law $k^{-1}$, where a maximum local peak occurs in the low wave number region of the $M2$ case, we decided to select $k_x h/2 \approx 0.5$ in order to separate low wave numbers from the high wave number regime. The wave number for which the beginning of the power law occurs in the spectra was observed to decrease with an increasing diameter of the roughness elements. In the $SP1$ case, the beginning of the power law was observed at $k_x h/2 \approx 0.4$, whereas the $SP2$ case resulted in a slightly lower wave number $k_x h/2 \approx 0.3$.

The area under the premultiplied spectrum curve can be interpreted as the corresponding energy levels at specific wave numbers [5]. Therefore, we have presented premultiplied spectra of the velocity fluctuations in Figure 9b,d,f in order to study the energy contents with corresponding wave numbers. A local maximum in the premultiplied spectra in the low wave number range is visible for all of the cases. The energy content of low wave number peaks for the $M2$, $SP1$, and $SP2$ cases resulted in $S_u(k_x h)/2u_\tau^2 = 0.213$, $0.175$, and $0.179$, respectively. Thus, the results indicate that an increasing particle height in the channel decreases the strength of the VLSM, which may even suppress the evolution of the VLSM of coherent structures due to the presence of a very large roughness height as compared to the boundary layer thickness which influences the logarithmic layer [9]. On the other hand, high wave number fluctuations were observed to decrease with increasing diameter of the roughness elements.

Spectrum and premultiplied spectrum analysis of the drag force was conducted in order to understand the effect of VLSM and LSM of coherent structures on the hydrodynamic forces. The $M2$ and $SP1$ cases resulted in a bi-modal distribution in the premultiplied spectra, as visible in Figure 10. The local maxima within the low wave number range resulted in a significant decrease with increasing roughness element height or decreasing particle exposure, although premultiplied spectra of the velocity fluctuations of the $M2$, $SP1$ and $SP2$ cases exhibited similar local maximum values. Therefore, our findings are in line with previous investigations [14,36]. The influence of the VLSM on the $SP2$ case is negligible. The area under the high wave number premultiplied spectrum curve of the fluctuations of the drag force corresponding to the LSM of the coherent structures was observed to decrease with an increasing roughness element height, although the magnitude of the local maximum was observed to be independant of the protrusion level, wxcept for the $M2$ case. The investigation conducted by Cameron et al. [14] reported that the high wave number peaks are due to the influence of the pressure field in the vicity of the particle. Our results lend to support the same conclusion, because hydrodynamic forces

for varying exposure levels showed that the mean drag force decreases with decreasing particle exposure.

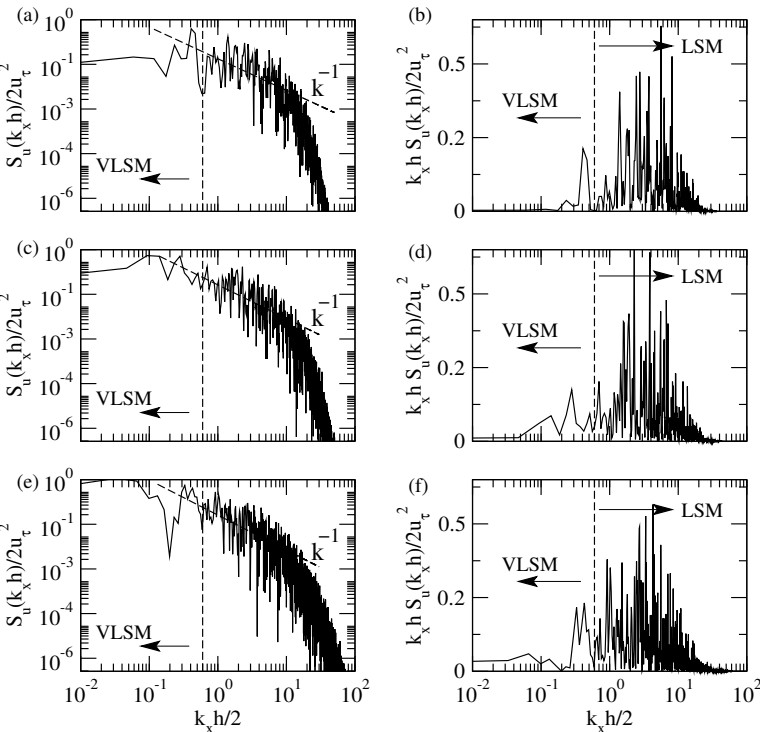

**Figure 9.** Energy spectrum analysis of the velocity fluctuations; (**a**–**f**) correspond to the *M*2, *SP*1 and *SP*2 cases, respectively; (**a**,**c**,**e**) depict energy spectra, whereas (**b**,**d**,**f**) depict the premultiplied energy spectra.

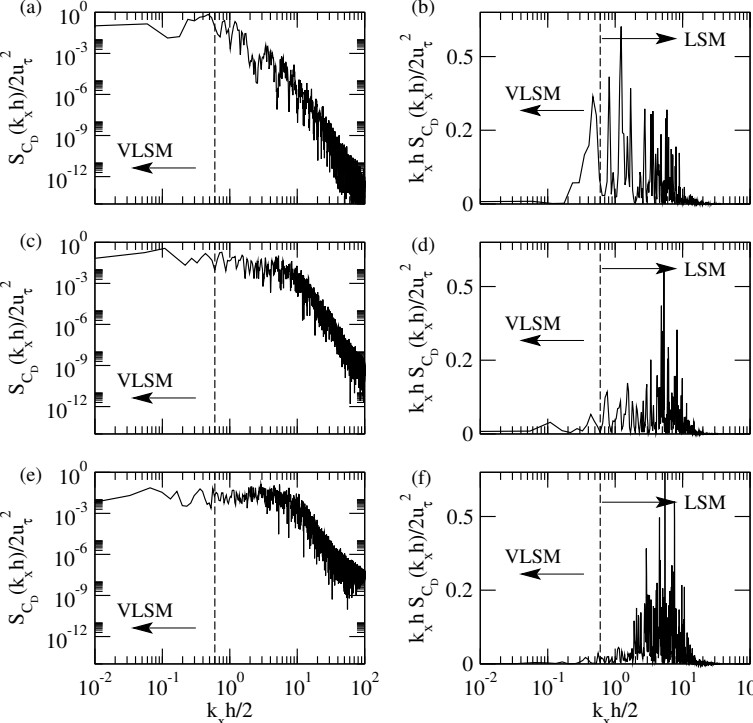

**Figure 10.** Energy spectrum analysis of the drag force; (**a**–**f**) correspond to the *M*2, *SP*1 and *SP*2 cases, respectively; (**a**,**c**,**e**) depict energy spectra, whereas (**b**,**d**,**f**) depict premultiplied energy spectra.

Figure 11 depicts the spectrum of the lift force and its corresponding premultiplied spectrum. The premultiplied spectra of the lift force of the fully exposed particle ($M2$) resulted in a bi-modal distribution, which was similar to the premultiplied spectra of the drag force, exhibiting a weak local maximum within the low wave number region at $k_x h/2 \approx 0.5$. Cases $SP1$ and $SP2$ only exhibited a uni-modal distribution within the low wave number range. The energy contents of the local maximum of the high wave number fluctuations significantly decreased with decreasing particle exposure.

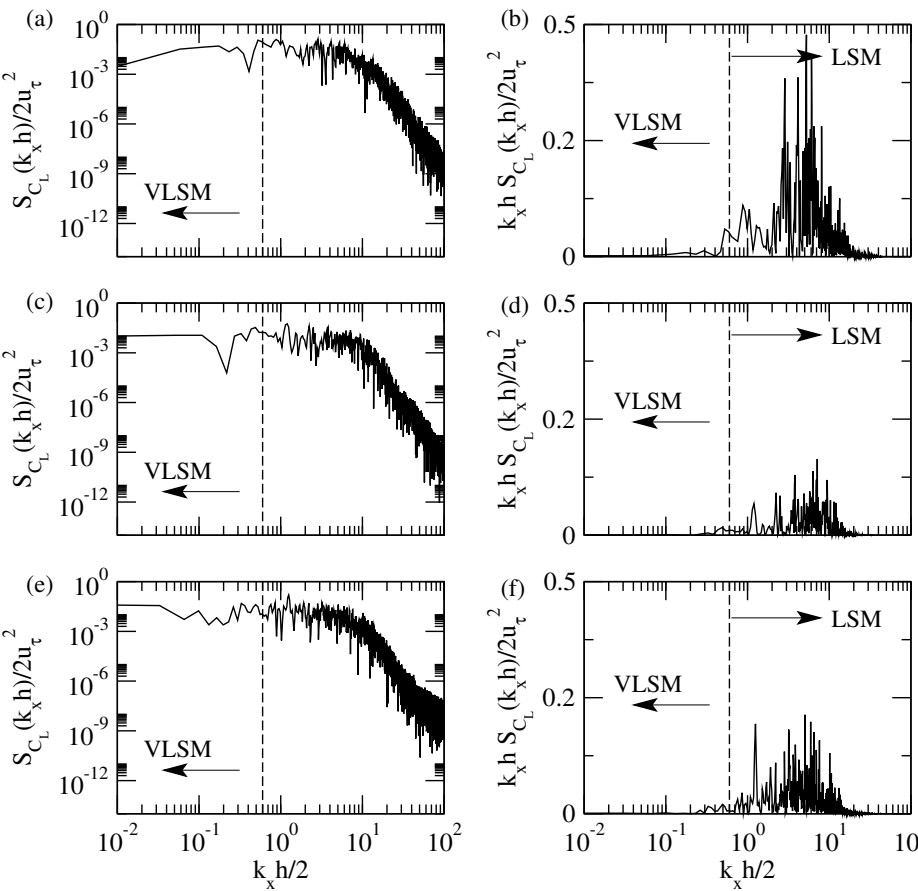

**Figure 11.** Energy spectrum analysis of the lift force; (**a**–**f**) correspond to the $M2$, $SP1$ and $SP2$ cases, respectively; (**a**,**c**,**e**) depict energy spectra, whereas (**b**,**d**,**f**) depict the premultiplied energy spectra.

## 4. Conclusions

Large Eddy Simulation of an open channel flow with two varieties of rough-wall boundary conditions has been carried out. The results were compared to the literature results of Yücesan et al. (2021), *Journal of Hydraulic Research*, for a single sediment particle that was mounted on a smooth-wall in fully developed turbulent open channel flow. Mean drag and lift forces were calculated and it was observed that the hydrodynamic forces decrease in magnitude with decreasing particle exposure, whereas the coefficient of variation increased. Thus, the rate of change in the mean hydrodynamic force is greater than the standard deviation and, consequentially, force fluctuations along the streamwise and wall-normal directions become more significant due to the increasing shielding effect.

The auto-correlation functions of the hydrodynamic forces were investigated and the drag force was observed to decay faster with decreasing particle exposure, while the rate of decay of the auto-correlation function of the lift force was observed to be almost independent of the particle exposure. Correlations between drag and lift forces were computed, and it was found that the cross-correlation function resulted in a higher coefficient for a fully exposed particle when compared to a half-exposed particle.

The premultiplied spectra of the velocity fluctuations were studied for the *M*2, *SP*1, and *SP*2 cases, and two modes of spectral peaks were identified: a low wave number peak indicating the presence of VLSM in the turbulent open channel and a high wave number peak indicating the influence of the LSM of the coherent structures. The local maxima within the low wave number regimes for the rough-wall cases were observed to be smaller in magnitude as compared to the smooth-wall case. Premultiplied spectra of the drag force showed that the fully exposed particle was characterised by a bi-modal shape, which was composed of peaks of a low wave number and high wave number. Similarly, the half-exposed particle exhibited peaks of a weak low wave number and a high wave number. The energy content of the fluctuations of the drag force in the premultiplied spectra were observed to decrease with decreasing exposure, despite that the magnitude of the premultiplied spectral peaks in the *M*2 and *SP*1 cases were determined to be identical. The fully hidden particle in the *SP*2 case was observed to be unaffected by the VLSM, which may be due to the shielding effect that yields no influence on the drag force. Fluctuations of the lift force on a smooth-wall reveal the existence of a weak local maximum within the low wave number region and a high amplitude peak at high wave numbers. Thus, fluctuations of the lift force of a single particle mounted on a smooth-wall were identified to possess a bi-modal distribution. However, this behaviour was not observed for the *SP*1 or *SP*2 cases. These results indicate that VLSM may not have an influence on the lift force of a particle on a rough-wall. The energy content of the premultiplied spectra of the fluctuations of the lift force within the high wave number region exhibits a decreasing trend for the rough-wall cases in comparison to the smooth-wall boundary. The influence of the LSM of the coherent structures gives an explanation for this observation, which may influence the pressure field serving as the responsible mechanism in the vicinity of the particle.

**Author Contributions:** Conceptualization: S.Y., C.H. and M.T.; Data curation: S.Y.; Formal analysis: S.Y.; Funding acquisition: C.H., H.H. and M.T.; Investigation: S.Y.; Methodology: S.Y.; Project administration S.Y., C.S., C.H., H.H. and M.T.; Resources: S.Y., C.S., C.H., H.H. and M.T.; Software: S.Y., P.G.; Supervision: C.H., H.H. and M.T.; Validation: S.Y., D.W., P.G., J.S., C.S., C.H., H.H. and M.T.; Visualization: S.Y.; Writing—original draft: S.Y.; Writing—review & editing: S.Y., D.W., J.S., C.S., C.H., H.H. and M.T.; All authors have read and agreed to the published version of the manuscript.

**Funding:** The financial support by the Christian Doppler Research Association, the Austrian Federal Ministry for Digital and Economic Affairs and the Austrian National Foundation of Research, Technology and Development is gratefully acknowledged.

**Institutional Review Board Statement:** Not applicable.

**Informed Consent Statement:** Not applicable.

**Data Availability Statement:** The data presented in this study are available on request from the corresponding author.

**Acknowledgments:** The computational results presented have been achieved using the Vienna Scientific Cluster (VSC).

**Conflicts of Interest:** The authors declare no conflict of interest. The funders had no role in the design of the study; in the collection, analyses, or interpretation of data; in the writing of the manuscript, or in the decision to publish the results.

## Abbreviations

The following abbreviations are used in this manuscript:

| | |
|---|---|
| LSM | Large-Scale Motion |
| TKE | Turbulent Kinetic Energy |
| VLSM | Very-Large-Scale Motion |

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
