# Peer review of "Interaction of Very Large Scale Motion of Coherent Structures with Sediment Particle Exposure"

_water, doi:10.3390/w13030248_

Round 1

Reviewer 1 Report

The authors have addressed my concerns about the original manuscript properly. I would like to recommend its publication in Water. 

Author Response

We would like to thank you for the reviewer's comments during the revision round 1.

Reviewer 2 Report

This is my second review of the manuscript that addresses the impact of large scale turbulent motion on the hydrodynamic forces acting on particles mounted to a wall. This is done by means of large eddy simulations using the CFD toolbox OpenFoam. Even though I liked the scope of the study, I had to reject the manuscript in my first review, because of substantial doubts about the methods used. This decision was based on two major concerns regarding the computational setup. The authors have now provided a substantially revised version that has improved the overall quality of the manuscript by quite a bit. Figures 1 and 3 of the original submission were improved to better understand the setup and two new figures were added to discuss the total domain size by means of two-point correlations and to visualize the coherent structures. However, I still have concerns about the way the simulations have been carried out. Most importantly, the second major concern of my first review on the impact of the downstream boundary on the simulation results was not addressed in a quantitative manner. Without this assessment, I am afraid I can still not recommend the manuscript for publication.

In particular, I think the following issues have not been resolved, yet:

1) The authors claim that they have applied a free-surface condition at the top wall / surface, but this would yield a velocity profile of the streamwise condition that comes out to be orthogonal to the top wall. This is clearly not the case in figures 1 and 3. This makes me wonder what kind of boundary condition was actually used for the top wall in this study?

2) In my first review, I had concerns about the impact of the downstream boundary condition on the simulation results. The authors now provide a better visual representation of the geometry and claim based on the qualitative inspection of the figures that there is no influence. This may be true, but it is not a valid proof to refute my concern. Instead, quantitative proof is needed to rule out this possibility. In my mind, the only way to show this would be running additional simulations with an extended downstream part to show that, in fact, the results do not change with increasing domain size. Hence, my concern from the first round of review still stands. Since this concern could potentially render the simulation results inadequate for the interpretation of drag forces on the particle, it will be crucial to resolve this issue properly.

3) I hope that the authors know and understand that the effective viscosity is not a physical meaningful value to use for the computation of a Reynolds number. It is used as a model to express an excess momentum diffusion to account for mixing that is not resolved by the numerical grid. On the contrary, it can by no means be used to express the balance of inertial and viscous forces in a viscous flow, i.e. the Reynolds number. I would therefore advice to remove this definition as it raises substantial doubts about the authors’ understanding of this essential physical parameter.

And a minor comment: l. 120: replace “grid density” with “grid resolution”

Round 2

Reviewer 2 Report

The authors have successfully addressed the concerns #1 and #3 of my previous review. Unfortunately, they were not able to answer concern #2 in a quantitative manner, which is due to constraints in computational resources available to the author. However, I believe that a brief discussion of the effect of the limited downstream domain size of the computational domain maybe a fair addition to raise awareness that this can indeed be a critical issue. I also think that authors should make it clear in their sketch shown in Figure 1 that there exists a free-slip condition at the top boundary to avoid confusion. However, these will only be minor changes and additions to the present manuscript which can be dealt with in a straightforward manner during the production process. Hence, I am happy to recommend the manuscript for publication.

Author Response

Thank you for the positive assessment and the opportunity to resubmit a revised version of the manuscript. According to the suggestion by the reviewer, we amended Figure 1 to show the free-surface boundary condition and added the following text in section 2.1 of the manuscript: 'The outlet is located ∼ 5.8d (or ∼ 7308(ν / uτ) in viscous units) downstream of the center of volume of the particle. While this is expected to be large enough to avoid any influence of the outlet boundary condition on the hydrodynamic forces acting on the particle, a further downstream extension of the domain would have been desirable, but proved computationally unfeasible.

This manuscript is a resubmission of an earlier submission. The following is a list of the peer review reports and author responses from that submission.

Round 1

Reviewer 1 Report

The present study uses Large Eddy Simulations to compute the hydrodynamic forces on single particles mounted on a rough wall. Two scenarios are considered, one with 50% protrusion (SP1) and one with zero protrusion (SP2) into the flow, and the results are compared to a previous study that was not accessible at the time of the review as it is apparently still under review. The overarching aim of the research was to identify the role of large-scale turbulent motion on the interaction of the turbulent flow with the sediment bed. This research question is interesting and deserves to be investigated. Unfortunately, I have two major concerns that both deal with the simulation setup and need to be addressed, because they can both have implications on the simulation results essentially rendering them to be inadequate for the analysis presented here. Hence, even though I like the general objective of the study, I cannot recommend the manuscript for publication. I am voting “reject” for the simple reason that it may take too long to conduct additional simulations (or in the worst case redo the entire study) within the time constraints given by Water. However, if the authors have already considered the issues outlined below, then I would be happy to recommend a Major Revision.

1) It is stated on line 105 that the particles were place at x/d=57.7 and x/d=28.85, for SP1 and SP2 respectively, where x is the streamwise coordinate and d=0.026 m is the diameter of the particle under investigation. Furthermore, it is stated the the upstream boundary condition at x=0 was an inflow generated from a precursor simulation of a recirculated turbulent open channel flow. It is clear that feeding this precursor flow into the main simulation domain will create some transition zone, where the flow adjusts from a smooth to a rough surface. Usually, best-practice experimental guidelines allow ten times the flow depth (h) to readjust the flow for a well-developed flow condition. That is a statistically stationary flow condition that does not change with the streamwise direction further downstream. Here, I can deduce that the particles were placed at x = 57.7 d = 1.5 m = 8.77h and x = 28.85 d = 0.75 m = 4.3h for SP1 and SP2, respectively, which are both well below the best-practice value of 10. Hence, I have strong doubts that the flow under investigation was well-developed. A plot showing delta, the boundary layer thickness, as a function of the streamwise coordinate is urgently missing. If there is still a developing boundary layer at the position of the mounted particle, than this study will forfeit its generality.

2) The downstream boundary condition with d u_i / d x = 0 is rather unusual. The default outflow condition for turbulent flows is the convective outflow condition. Furthermore, the investigated particles are place quite close to that outflow boundary. For SP1, the position even seems to coincide with the domain boundary. Consequently, the authors need to clarify the impact of the downstream boundary condition on their simulation results by conducting further simulations with longer streamwise extent to determine the extent needed to obtain results that are independent of this choice. If there still is an influence from the downstream boundary, then the simulation results are not to be interpreted in a physical sense.

Apart from these major issues, I found quite a few minor issues as well:

1) l. 19: “Further investigations” → which ones?

2) l. 33 and throughout the article: “Chan et al.” should be Chan-Braun et al.

3) l. 32-47: What are the Reynolds numbers used in those cited studies and how do they compare to the present study

4) l. 58: “not affected”

5) l. 67: “resulted in an increase with increasing...” → increase of what?

6) l. 70-71: “Today, our understanding … has been established” → how exactly is it established?

7) l. 88: Why is there a distance of Delta p needed?

8) figure 2 and table 1: U_b or u_b?

9) Figure 2: Where in the channel were the data gathered? Is u’_x time-averaged or instantaneous? The caption of that figure does not refer to (b).

10) Table 1: Was U taken from the precursor simulation? How was tau_w defined? The Reynolds numbers should not be computed with the effective viscosity.

11) l. 116-117: “incompressible … equations” → equations cannot be incompressible, but they can describe an incompressible fluid

12) Axes on figure 3 should reflect the streamwise and vertical coordinates. Moreover, this seems to be a local coordinate system that is centered on the particle that was investigated. The figure seems to be a zoom into the domain. Also, the particle location does seem to coincide with the values given in section 2.1 (l. 105). Figure 3a is revealing: One can see that the wake of the large particle is influenced by the downstream boundary.

13) l. 188: this statement sounds too conclusive and is strictly speaking applicable to the presented setup only.

14) figure 7-9: Why not show the entire spectrum all the way down to the grid scale?

15) l. 299-300: This is a conjecture that is not backed up by the data.

16) l. 309: “hydrodynamic forces decrease ...with particle exposure” → this is not a new finding and references should be given

17) l. 334-335: “An explanation for this observation is given by the fluctuations of the pressure field” → but the pressure field is not discussed.

Reviewer 2 Report

This manuscript presents the numerical results of an open channel flow with rough wall boundary using LES, focusing on the two types of three-dimensional coherent structures, namely LSM and VLSM. The simulations appear carefully planned conducted and the results satisfactorily presented. However, there are several issues needed to be addressed before the manuscript could be recommended for publication.

  1. Regarding the numerical scheme, has mesh independence test been conducted?
  2. Since the study is focused on two the types of three-dimensional coherent structures, namely LSM and VLSM. It is therefore desirable to provide some representative snapshots, giving the reader a visual impression in terms of their origin, size and spatial development, etc.
  3. In Section 3.4. Some spectral plots show that the spectral peak within the low wave number region is rather vague. What is the reason? Is it because the size of discrete data samples is too small? Also, the authors need to justify why using the value of kxh/2=0.5 discriminate the scales of LSM and VLSM.